# Dual Oxidase 2 (DUOX2) as a Proteomic Biomarker for Predicting Treatment Response to Chemoradiation Therapy for Locally Advanced Rectal Cancer: Using High-Throughput Proteomic Analysis and Machine Learning Algorithm

**DOI:** 10.3390/ijms232112923

**Published:** 2022-10-26

**Authors:** Hyebin Lee, Han Suk Ryu, Hee Chul Park, Jeong Il Yu, Gyu Sang Yoo, Changhoon Choi, Heerim Nam, Jason Joon Bock Lee, In-Gu Do, Dohyun Han, Sang Yun Ha

**Affiliations:** 1Department of Radiation Oncology, Kangbuk Samsung Hospital, Sungkyunkwan University School of Medicine, Seoul 03181, Korea; 2Department of Pathology, Seoul National University Hospital, Seoul National University College of Medicine, Seoul 03080, Korea; 3Department of Radiation Oncology, Samsung Medical Center, Sungkyunkwan University School of Medicine, Seoul 06351, Korea; 4Department of Pathology, Kangbuk Samsung Hospital, Sungkyunkwan University School of Medicine, Seoul 03181, Korea; 5Proteomics Core Facility, Biomedical Research Institute, Seoul National University Hospital, Seoul 03080, Korea; 6Department of Pathology, Samsung Medical Center, Sungkyunkwan University School of Medicine, Seoul 06351, Korea

**Keywords:** rectal cancer, chemoradiation, predictive biomarker, proteomics, mass spectrometry

## Abstract

High-throughput mass-spectrometry-based quantitative proteomic analysis was performed using formalin-fixed, paraffin-embedded (FFPE) biopsy samples obtained before treatment from 13 patients with locally advanced rectal cancer (LARC), who were treated with concurrent chemoradiation therapy (CCRT) followed by surgery. Patients were divided into complete responder (CR) and non-complete responder (nCR) groups. Immunohistochemical (IHC) staining of 79 independent FFPE tissue samples was performed to validate the predictive ability of proteomic biomarker candidates. A total of 3637 proteins were identified, and the expression of 498 proteins was confirmed at significantly different levels (differentially expressed proteins—DEPs) between two groups. In Gene Ontology enrichment analyses, DEPs enriched in biological processes in the CR group included proteins linked to cytoskeletal organization, immune response processes, and vesicle-associated protein transport processes, whereas DEPs in the nCR group were associated with biosynthesis, transcription, and translation processes. Dual oxidase 2 (DUOX2) was selected as the most predictive biomarker in machine learning algorithm analysis. Further IHC validation ultimately confirmed DUOX2 as a potential biomarker for predicting the response of nCR to CCRT. In conclusion, this study suggests that the treatment response to RT may be affected by the pre-treatment tumor microenvironment. DUOX2 is a potential biomarker for the early prediction of nCR after CCRT.

## 1. Introduction

Neoadjuvant radiation therapy with concurrent chemotherapy (CCRT) followed by total mesorectal excision (TME) is considered the standard treatment for locally advanced rectal cancer (LARC). This multidisciplinary team approach has improved local tumor control and the overall survival of patients with rectal cancer [1,2]. Although radical surgery is the mainstay of treatment for rectal cancer, it is associated with significant morbidity that permanently impairs the quality of life of patients, particularly in patients with distal tumors requiring low anastomoses or a permanent colostomy [3,4]. In this regard, there has been a growing interest in organ-preservation strategies, in which surgery is omitted, or at least deferred until local tumor progression, in patients who have achieved a complete pathologic response (pCR) after CCRT. According to previous studies, 10–25% of all LARC patients show pCR after neoadjuvant CCRT [2,5]. Several recent studies confirmed that patients with pCR after CCRT have better long-term, disease-free, and overall survival than those without pCR [5,6]. In this context, the non-operative management (NOM) of rectal cancer, also known as the “watch-and-wait” strategy, has been actively investigated [7,8,9]. NOM can be adopted for selected rectal cancer patients who achieve a complete clinical response (cCR), which is defined as the clinical absence of residual tumor assessed by a digital rectal examination and endoscopic and radiological surveillance. However, due to a significant discrepancy between cCR and pCR, there is a clear need to predict a pathologic response to CCRT. Several studies have been conducted to identify genetic and molecular biomarkers to predict the response to CCRT [10,11,12]. Despite these efforts, results are controversial and inconclusive, and no specific molecular markers have been definitively proven to be predictive.

Recent rapid advances in high-throughput sequencing and proteomic techniques with sophisticated computational algorithms have enabled highly quantitative analyses and deep profiling of clinical samples [13,14]. Formalin-fixed, paraffin-embedded (FFPE) tissue specimens are the most widely used samples in clinical diagnosis and cancer research because they provide not only economic advantages for longitudinal tissue specimen storage but also ensure a high degree of protein preservation. Due to the technical advances in protein extraction from FFPE samples for proteomic analysis, in-depth proteomic analysis using these samples leads to a more comprehensive understanding of functional tumor biology.

In this study, to identify novel protein biomarkers for predicting the response to CCRT in patients with LACR, in-depth proteomics based on high-resolution mass spectrometry (MS) was employed using FFPE specimens, which was obtained by endoscopic biopsy before the initiation of treatment. To discover suitable biomarkers, an integrative workflow was designed, including a machine learning algorithm, followed by immunostaining validation using an independent cohort.

## 2. Results

### 2.1. Overall Protein Identification and Quantification

The detailed clinicopathological features of training and validation sets are described in Table 1.

A large-scale proteome profiling of rectal adenocarcinoma tissue was performed based on CCRT response using endoscopic biopsy-derived FFPE samples. In total, 4345 and 3637 protein groups were identified and quantified at the 1% FDR level by a single-shot proteomic analysis of the FFPE samples, respectively (Figure 1A,B). In each case, the average number of identified and quantified proteins was more than 3000 proteins per sample, spanning eight orders of magnitude of MS signal intensity (Appendix A). After implementing a normalization process to correct for systematic bias across comparison groups, label-free quantification and two-sided *t*-tests yielded 498 differentially expressed proteins (DEPs) between the two groups—269 in the nCR group and 229 in the CR group—with a *p*-value < 0.05 after filtering fold change (Figure 1A). Hierarchical clustering based on proteomic expression resulted in two groups (Figure 1C). Principal component analysis (PCA) also revealed a tight clustering of two groups and their corresponding biological replicates, indicating distinct protein expression patterns within each group (Figure 1D).

A further two-group analysis between CR and nCR also demonstrated the overexpression of proteins that play a tumor-suppressive role, including AKAP12 [15], DCTN3 [16], and SELENOH [17], in the CR group (Figure 2). In the nCR group, several key proteins were upregulated, including DUOX2 [18,19], PQCTL [20], DIAPH1 [21], SUPT5H [22], and MUC13 [23], which promote cell migration and invasiveness (DIAPH1, SUPT5H, and MUC13), immune surveillance escape (QPCTL), and epithelial–mesenchymal transition (EMT; DUOX2) (Figure 2).

### 2.2. Characterization of Enriched Pathways Based on CCRT Response

To compare DEPs between the two groups, a Gene Ontology (GO) enrichment analysis in each protein cluster was performed (Figure 3). A GO analysis on the biological process revealed enrichment in the pathway related to organic acid metabolism, oxidation–reduction, transcription, and translation in the nCR group (Figure 3A,B). On the other hand, cytoskeleton organization, the regulation of cell motility, and multiple immune response-related processes, including antigen processing and presentation, cellular response to interferon gamma, vesicle organization and secretion, and endothelial cell development, were significantly enriched in the CR group (Figure 3A,C).

GO enrichment of top-ranked proteins displaying large magnitude changes or being statistically significant revealed ten different gene sets in the nCR group: organic acid metabolism with AMP1, BPNT1, and QARS1; oxidation–reduction process with DUOX2, IDE, NDUFA, IDH1, and IDH2; and transcription and translation with SUPT5H, AIMP1, QARS1, and MRPL. Another pathway enrichment analysis of top-ranked proteins from the CR group selected each set of proteins based on the statistical significance or intensity ratio as follows: TPM1 and DCTN3 were categorized into cytoskeleton organization; AKAP12, SERPINF1, and MACAM were categorized into cell motility regulation; HLA-B, HLA-C, and HLA-F were categorized into antigen processing and presentation processes; HLA-DQB1, HLA-DRB1, RAB12, and MRC1 were categorized into cellular response to interferon gamma; and A1BG, STAM2, MYH10, and SNX3 were categorized into vesicle organization.

In terms of subcellular distribution of DEPs according to CCRT response, GO analysis of cellular components showed a significantly different localization of abundant proteomes between these two groups. As shown in Figure 3D, numerous proteins in the nCR group were mainly localized in the mitochondria, which generate most of the chemical energy needed to power biochemical reactions in cells. DEPs in the nCR groups were also largely localized in the ribosomes, which serve as the sites of protein synthesis by linking amino acids with mRNA molecules. In contrast, in the CR group, DEPs were categorized into cytoskeleton organization, cell junctions, and extracellular complexes and involved in regulating cellular structures and proteins localized in the vesicles and organelle envelopes that control the transport of substances (Figure 3D).

### 2.3. Machine Learning Approach for Selecting Protein Biomarkers to Predict Treatment Response

To evaluate the predictive power of individual proteins with respect to CCRT response, a machine learning approach was adopted in which an algorithm, SVM with RBF kernel, was applied. In this analysis, DUXO2 was ranked as the top protein, whose higher expression level is best classified between the nCR and CR groups. (Figure 4A).

### 2.4. Immunohistochemical Validation of the Predictive Value of DUOX2

DUOX2 was evaluated in the independent validation set of an independent 79-patient cohort using IHC analysis. All enrolled cases were classified into two groups, nCR (n = 62) and CR (n = 17). Sequential surgical samples were available and used by the pathologists to examine the presence of residual tumor cells. All tissues were examined blindly without any clinicopathological information for a more accurate evaluation. Protein expressions for the antibody varied according to the score created on a scale of 0 to 3 using the H-score. Nonparametric analysis with a Mann–Whitney U test indicated the association of high levels of DUOX2 with resistance to CCRT (*p* = 0.044; Figure 4B).

## 3. Discussion

The present study using pre-treatment FFPE biopsy specimens of rectal adenocarcinoma demonstrated the distinct expression of proteomes between the CR and nCR groups after CCRT in LARC patients. The major proteomes that were abundant in the CR group were linked to cytoskeleton organization, immune response process, and a vesicle-associated protein transport system. On the contrary, proteins associated with biosynthesis, transcription, and translation processes were mainly expressed in the nCR group. These findings indicate that the tumor microenvironment prior to treatment can affect the antitumor treatment response. DUOX2 was prioritized as a protein biomarker for the prediction of nCR because it consistently ranked first in various statistical analyses, including feature selection analysis, which adopted a machine learning algorithm. The validation of these results through IHC analysis provided preliminary confirmation that the overexpression of DUOX2 in rectal cancer is associated with treatment resistance.

Another noteworthy finding is the abundance of immune-related proteomes in the CR group, including interferon gamma (IFNG)-related proteins and adaptive-immunity-associated proteins (Figure 3). Immune infiltrated tumors, so-called immunologically “hot” tumors, are known to be more responsive to immunotherapy [24]. Even though the possible association between the immunologic aspect of the tumor microenvironment and treatment response is beyond the scope of the current study, further studies regarding this might be beneficial for the management of LARC.

DUOX2 is a member of the NADPH oxidase (NOX) family (NOX1-5, and DUOX1-2). It is the one of the main sources of endogenous reactive oxygen species (ROS), which are increased following cell exposure to internal or external stimuli [25]. DUOX2 is known to play an important role in the innate host defense against pathogens in the rectal mucosal epithelia via producing endogenous ROS [26]. Increased ROS levels, due to imbalances between ROS production and cellular responses to counteract their actions, is one of the hallmarks of cancer. ROS accumulation and the resulting oxidative stress contribute to carcinogenesis, tumor aggressiveness, and antitumor treatment resistance by causing DNA damage and changing intracellular signaling pathways via post-translational modifications [27,28,29]. Thus, it can be hypothesized that DUOX2-dependent oxidative stress affects treatment outcomes in rectal cancer patients treated with CCRT. Although there have been in vitro studies that demonstrate the role of DUOX2 in carcinogenesis [30], tumor progression [18], and chemotherapeutic resistance [19], particularly in colorectal and pancreatic adenocarcinoma, the association between DUOX2 and CCRT response has rarely been reported.

Although there is limited evidence to explain the association between DUOX2 and CCRT response in rectal cancer patients, the findings of the present study seem to agree with those of other transcriptomic studies. Lin et al. conducted data mining in a public transcriptome database (GSE35452) comprising 46 rectal cancer patients who received preoperative CCRT and radical surgery [31]. They found that DUOX2 was the most significantly upregulated transcript in the non-responder group.

To the best of our knowledge, this is the first study to report DUOX2 as a potential predictive biomarker of treatment resistance after CCRT for rectal cancer at the proteomic level. In addition, the current study presents a new methodology in radiation oncology research. Quantitative proteomic analysis using FFPE specimens has begun to emerge in biomarker studies over the past few years. This approach will provide more insights into the molecular mechanism of radiation sensitivity and resistance.

The detection of DUOX2 in human tissue remains an unsolved issue in this study. Although significantly different expressions between the CR and non-CR groups were observed in IHC, sole IHC seems to be insufficient for use in clinical settings. Like in the case of HER2 (human epidermal growth factor receptor-2) tests, such as in situ hybridization, another molecular assay for enhancing the detection accuracy of DUOX2 may need to be developed.

Despite these limitations, the findings of the present study suggest that DUOX2 can be a novel biomarker for predicting non-CR after CCRT in patients with rectal cancer, highlighting the importance of the tumor microenvironment with respect to treatment responsiveness.

## 4. Materials and Methods

### 4.1. Sample Collection

We retrospectively reviewed the medical records of LARC patients who were treated with neoadjuvant CCRT followed by radical surgery between January 2012 and October 2016. The inclusion criteria for this study were as follows: (1) histologically confirmed adenocarcinoma of the rectum; (2) no history of prior chemotherapy or radiation therapy; (3) clinical T3 disease with regional lymph node metastasis and no distant metastasis at diagnosis; (4) patients who completed CCRT and underwent TME 7 to 9 weeks after RT completion; and (5) available pre-treatment endoscopic biopsy and post-CCRT surgical specimens.

The treatment consisted of neoadjuvant CCRT followed by radical surgery. Radiation therapy (RT) was delivered to the whole pelvis at a total dose of 44 Gy in 22 fractions, 5 days per week. Concurrent chemotherapy was administered with fluoropyrimidine-based regimens. All patients underwent TME eight weeks after the completion of CCRT. The pathologic response to CCRT was assessed by a pathologist and was reported as a grade based on resection specimens. Tumor regression grade (TRG) was assessed according to the Dworak’s TRG system. Grades were defined as follows: grade 0, no response; grade 1, dominant tumor mass with obvious fibrosis, vasculopathy, or both (minimal response); grade 2, dominant fibrotic changes with a few easy-to-find tumor cells or groups (moderate response); grade 3, few tumor cells in fibrotic tissue with or without mucous substance (near complete response); grade 4, no viable tumor (complete response). For proteomic analysis, the cases were dichotomized into complete responder (CR) and non-complete responder (nCR) groups; the CR group indicated patients with TRG 4, whereas the nCR group comprised patients with TRG 0 to 3.

A total of 13 FFPE samples in the training set and an independent cohort of 79 samples, obtained by endoscopic biopsy before treatment, were employed for quantitative proteomic analysis and validation using immunohistochemistry (IHC) of selected proteomic biomarkers, respectively.

### 4.2. LC-MS/MS Proteomics Analysis

Figure 5 indicates a brief overview of our approach for the proteomic discovery of novel biomarkers. Briefly, tumor cells from unstained slides were scraped, and the peptide was digested using the filter-aided sample preparation (FASP) procedure as previously described [32]. Desalted pooled peptides were fractionated using the stage-tip-based high-pH peptide fractionation method [33], then a liquid chromatography–tandem mass spectrometry (LC-MS/MS) analysis was performed using a Q Exactive Plus Hybrid Quadrupole-Orbitrap mass spectrometer (Thermo Fisher Scientific Inc., Waltham, MA, USA), coupled to an Ultimate 3000 RSLC system (Dionex, Sunnyvale, CA, USA) via a nanoelectrospray source, as previously described [33]. Peptide samples were separated on a two-column system, consisting of a trap column and an analytic column (75 μm × 50 cm) with a gradient from 7% to 32% ACN applied within a period of 120 min at 300 nL/min and analyzed by MS. Survey scans (350 to 1650 *m*/*z*) were acquired at a resolution of 70,000 at *m*/*z* 200. MS/MS spectra were acquired at an HCD-normalized collision energy of 30 with a resolution of 17,500 at *m*/*z* 200. The maximum number of ion injections for the full and MS/MS scans was 20 and 100 ms, respectively.

### 4.3. Data Analysis and Peptide Identification, Label-Free Quantification

Mass spectra were processed using MaxQuant version 1.5.3.1 [34]. MS/MS spectra were searched in the Human UniProt protein sequence database (December 2014, 88,657 entries) using the Andromeda search engine with a 6 ppm precursor ion tolerance for total protein level analysis [35]. Primary searches were performed using MS/MS ion tolerance and set to 20 ppm. Cysteine carbamidomethylation N-acetylation of protein and oxidation of methionine were set as fixed and variable modifications, respectively. Enzyme specificity was set to complete tryptic digestion. Peptides with a minimum length of six amino acids and up to two missed cleavages were considered. The required false discovery rate (FDR) was set to 1% at the peptide, protein, and modification levels. We enabled the “Match between Runs” option on the MaxQuant platform to maximize the number of quantification events across samples.

For label-free quantification, the intensity-based absolute quantification (iBAQ) algorithm [36] was used as a part of the MaxQuant platform. Briefly, iBAQ values were calculated using MaxQuant as raw intensities divided by the number of theoretical peptides. Thus, iBAQ values are proportional to the molar quantities of the proteins. All statistical analyses were performed using Perseus software [37]. Missing values were imputed based on a standard distribution (width = 0.15, downshift = 1.8) to simulate signals for proteins of low abundance. Finally, data were normalized using width adjustment, which subtracts medians and scales for all values in a sample to show equal interquartile ranges (Appendix A). Pairwise comparison of the proteomes’ two-sided t-tests was performed by utilizing the threshold *p*-value and a significance level of 5%. A protein was considered statistically significant if its fold change was ≥1.5 and *p*-value was ≤0.05.

All proteomic datasets were submitted to the ProteomeXchange Consortium via the PRIDE (http://proteomecentral.proteomechange.org) (accessed on 24 December 2020). partner repository with the dataset identifier PXD023302.

### 4.4. Machine Learning Analysis for Prioritizing Predictive Biomarkers

The determination of signature protein combinations utilizes the concept of recursive feature elimination. Since recursive feature elimination selects a variable subset via machine learning model performance, we employed the machine learning algorithm of a support vector machine (SVM) with radial basis function (RBF) kernel from the learning plug in Perseus to find a minimal signature that discriminated between the two groups. For classification, we used SVM with C10 for feature ranking and SVM with RBF Kernel and Sigma10. We performed leave-one-out cross-validation on the training set to classify samples between CR and nCR groups, thus creating a list of potential signatures with the highest accuracy scores.

### 4.5. Immunohistochemistry for Validation of Selected Biomarker

To validate the diagnostic utility of a proteomic biomarker (dual oxidase2—DUOX2), 79 independent FFPE tissue samples were examined using IHC. DUOX2 immunostaining was performed using Benchmark XT (Ventata Medical System, Inc., Tucson, AZ, USA). The monoclonal mouse anti-DUOX2 antibody (NB110-61576, Novus Biologicals, Centennial, CO, USA) was diluted to 1:100. The binding of the primary antibody was identified using an Optiview universal DAB kit (Ventana Medical Systems, Inc., Tucson, AZ, USA) according to the manufacturer’s protocol. IHC analysis results were interpreted by a semi-quantitative approach using an “H-score” [38] in a blind and independent manner by a pathologist (H.S.R.).

### 4.6. Bioinformatics and Statistical Analyses

Gene Ontology (GO) annotation was explicated using ToppGene Suite resources (https://toppgene.cchmc.org/) (accessed on 15 January 2021) [39]. Annotated MS/MS spectra were accessed through MS-Viewer [40]. For statistical evaluation of the H-score between the CR and nCR groups, we employed Kruskal–Wallis and Mann–Whitney U tests using the GraphPad Prism 8.0 program (GraphPad Software, Inc., San Diego, CA, USA). Student’s *t*-test was used to determine the extent of significance (*p*  ≤  0.05).

## 5. Conclusions

In conclusion, by using advanced proteomic techniques such as DUOX2, in the present study, we identified a novel promising predictive biomarker, DUXO2. The overexpression of DUOX2 can be used as an indicator of treatment-resistant tumors and can be applied as a new ancillary tool for the management of locally advanced rectal cancer. Further validation studies using larger sample sizes should be conducted to incorporate DUOX2 into clinical practice in the near future.

## Figures and Tables

**Figure 1 ijms-23-12923-f001:**
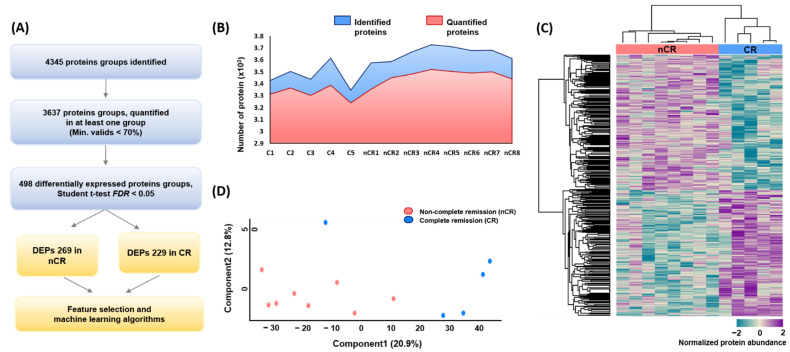
(**A**) Overall protein profiling. (**B**) Identified and quantified proteins among 13 rectal cancer biopsy samples. (**C**) Heatmap depicting unsupervised hierarchical clustering of 498 differentially expressed proteins (DEPs) between CR and nCR groups. (**D**) Principal component analysis (PCA) of proteome data from thirteen samples using the total set of proteins quantifies with an expression value across all samples.

**Figure 2 ijms-23-12923-f002:**
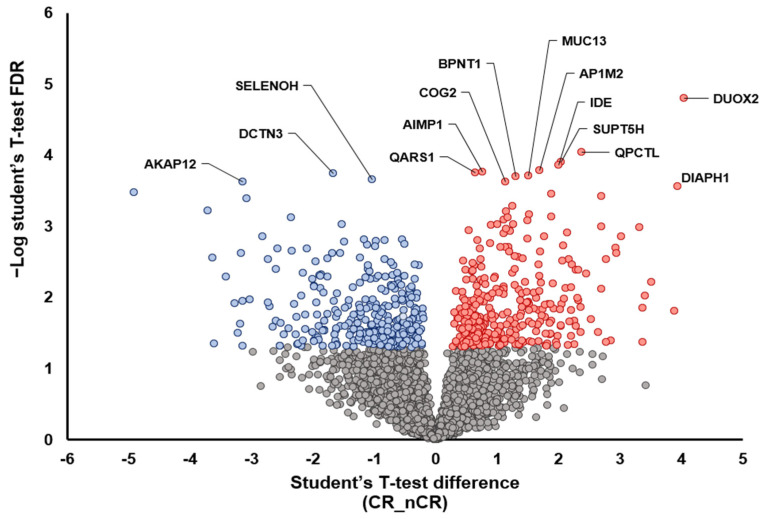
Volcano plot depicting the variance inexpression between CR and nCR groups. Highlighted data points indicate the key proteins that show the most significantly discriminative power (blue dots, increased intensity in CR group; red dots, increased intensity in nCR group).

**Figure 3 ijms-23-12923-f003:**
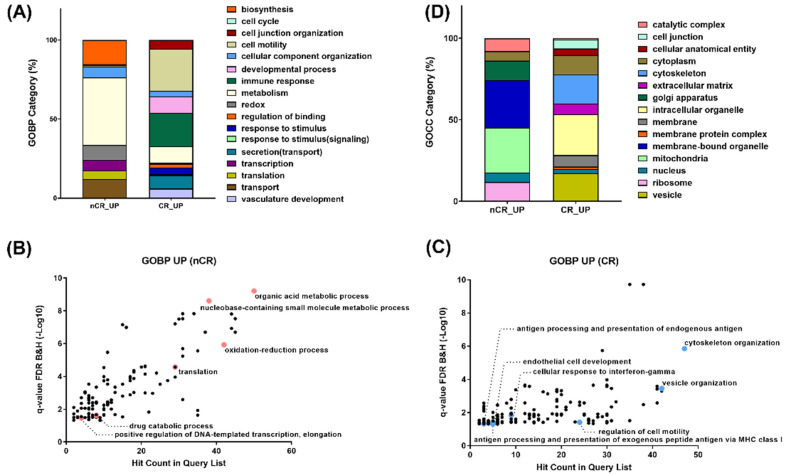
(**A**) Gene Ontology enrichment analyses on biological process on nCR (**B**) and CR (**C**) groups. (**D**) Gene Ontology enrichment analyses on cellular component.

**Figure 4 ijms-23-12923-f004:**
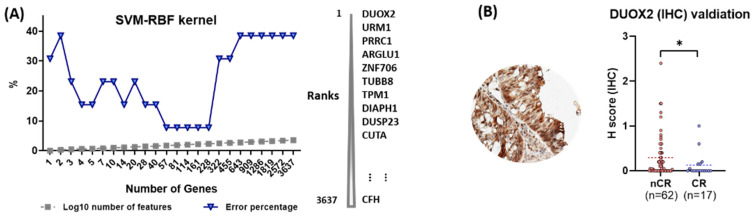
(**A**) Machine learning approaches to measure the predictive power of proteins for CCRT response. (**B**) Immunohistochemical (IHC) validation of dual oxidase 2 (DUOX2) as a predictive biomarker for CCRT response (statistical significance, * *p*-value < 0.05).

**Figure 5 ijms-23-12923-f005:**
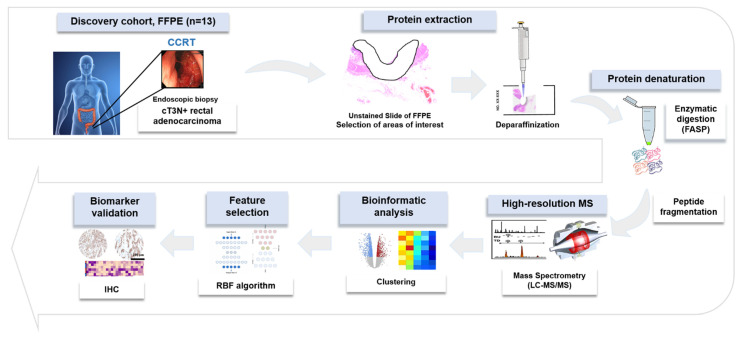
Overall experimental workflow.

**Table 1 ijms-23-12923-t001:** Patients and tumor characteristics of training (n = 13) and validation sets (n = 79).

		*Training Set (n = 13)*			*Validation Set (n = 79)*		
Characteristics		CR (n = 5), n (%)	Non-CR (n = 8), n (%)	*p*-Value	CR (n = 17), n (%)	Non-CR (n = 62), n (%)	*p*-Value
**Age**	≤60 years	2 (40.0%)	2 (25.0%)	0.569	10 (58.8%)	34 (54.8%)	0.770
	>60 years	3 (60.0%)	6 (75.0%)		7 (41.2%)	28 (45.2%)	
**Gender**	Male	3 (60.0%)	6 (75.0%)	0.569	11 (64.7%)	44 (71.0%)	0.619
	Female	2 (40.0%)	2 (25.0%)		6 (35.3%)	18 (29.0%)	
**Distance from anal verge**	≤5 cm	4 (80.0%)	6 (75.0%)	0.835	12 (70.6%)	45 (72.6%)	0.871
	>5 cm	1 (20.0%)	2 (25.0%)		5 (29.4%)	17 (27.4%)	
**Initial CEA level**	≤3.0 ng/mL	1 (20.0%)	3 (37.5%)	0.506	14 (82.4%)	26 (41.9%)	0.003
	>3.0 ng/mL	4 (80.0%)	5 (62.5%)		3 (17.6%)	36 (58.1%)	
**Type of Surgery**	LAR	5 (38.5%)	8 (61.5%)	N/A	16 (94.1%)	60 (96.8%)	0.612
	APR	–	–		1 (5.9%)	2 (3.2%)	

**CEA**, carcinomembryonic antigen; **LAR**, lower anterior resection; **APR**, abdominoperineal resection.

## Data Availability

All proteomic datasets were submitted to the ProteomeXchange Consortium via the PRIDE (http://proteomecentral.proteomechange.org) (accessed on 24 December 2020) partner repository with the dataset identifier PXD023302.

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
