# Peer review of "Dual Oxidase 2 (DUOX2) as a Proteomic Biomarker for Predicting Treatment Response to Chemoradiation Therapy for Locally Advanced Rectal Cancer: Using High-Throughput Proteomic Analysis and Machine Learning Algorithm"

_ijms, 2022, doi:10.3390/ijms232112923_

Round 1

Reviewer 1 Report

This study discovered a potential protein biomarker DUOX2 for prognosis of nCR to CCRR patients and kind of validated the transcriptomic data from previous study of another researcher that found this gene also was-upregulated. The methods workflow is sufficient, the result obtained are well interpret and discuss. But I have a few points that can be included mainly in the methodology 1) How many samples (replicates) used for proteomic analysis for nCR and CR group? In page 3, there is stated about ANOVA used to evaluate multigroup, but it does not mention in methodology, is it statistical analysis using Perseus??. Also, the LC-MS/MS method should be well described, add in all neccessary parameters for running and database search (can see example from proteomics journal). Section 4.3 and 4.6 overlap for the statistical analysis using Perseus, can be merged into one section. For discussion,authors can compare to other proteomics studies for rectal cancer in term of protein hits if this study is getting more protein hits? protein identified in this study similar to those other studies?

Author Response

This study discovered a potential protein biomarker DUOX2 for prognosis of nCR to CCRR patients and kind of validated the transcriptomic data from previous study of another researcher that found this gene also was-upregulated.

The methods workflow is sufficient, the result obtained are well interpret and discuss. But I have a few points that can be included mainly in the methodology

→ Thank you for your insightful review and comments. We made several corrections and clarifications in the manuscript after going over your comments. To expedite the processing of our revised manuscript, we added the page number and highlighted the changes in yellow in it. Again, we truly appreciate all your efforts to improve our manuscript.

[Comment 1] How many samples (replicates) used for proteomic analysis for nCR and CR group? In page 3, there is stated about ANOVA used to evaluate multigroup, but it does not mention in methodology, is it statistical analysis using Perseus?

[Response]: Thank you for your comment. We generated the proteomic data using MaxQuant and Perseus, which are the standardized software platform in interpreting protein quantification for biological and biomedical research (Tyanova. Nat Protocol 2016 and Tyanova et al. Nat. Methods 2016). We admit that there was a mistake in writing statistical method. We compared DEPs between two groups using t-test, but not ANOVA analysis, which was also performed using Perseus. We corrected this mistake about it, which were highlighted in skyblue color in page 3.

[Comment 2] Also, the LC-MS/MS method should be well described, add in all neccessary parameters for running and database search (can see example from proteomics journal).

[Response]: Thank you for your comment. As your comment, we added several sentences to supplement LC-MS/MS method, which were highlighted in yellow color.

[Comment 3] Section 4.3 and 4.6 overlap for the statistical analysis using Perseus, can be merged into one section.

[Response]: As your comment, we rearranged statistical analyses section, which were separated into proteomic identification (4.3) and other bioinformatics/immunohistochemical analyses (4.6).

[Comment 4] For discussion, authors can compare to other proteomics studies for rectal cancer in term of protein hits if this study is getting more protein hits? protein identified in this study similar to those other studies?

[Response]: As mentioned in Discussion section, we could not find out proteomic studies investigating predictive biomarkers to CCRT response for rectal cancer. Instead, transcriptomic study which showed same transcriptome (DUOX2) as a predictive biomarker was found, and we described their result in the Discussion.

Reviewer 2 Report

Table 1: The purpose of the percentage distribution is unclear. 

The sum of percentage points exceeds 100 percent.

Author Response

Table 1: The purpose of the percentage distribution is unclear. The sum of percentage points exceeds 100 percent.

[Response]: Thank you for your comment. While we provided percentage distribution to demonstrate that no significant difference existed between CR and non-CR group in most characteristics, we admit that there were a few mistakes in calculation and typographical errors with unclear message. Therefore, we modify the table as follows with the percentage distribution within CR and non-CR patients rather than from the entire population set. Please note that no significant difference in patient characteristics was observed between CR and non-CR group except initial CEA level.
